# Datasets, Documents, and Repetitions:
# The Practicalities of Unequal Data Quality

**Alex Fang**[1,2]    **Hadi Pouransari**[1]    **Matt Jordan**[3]
**Alexander Toshev**[1]    **Vaishaal Shankar**[4]    **Ludwig Schmidt**[2]    **Tom Gunter**[1]
[1]Apple    [2]Stanford    [3]work done while at UT Austin    [4]work done while at Apple

## Abstract

Data filtering has become a powerful tool for improving model performance while reducing computational cost. However, as large language model compute budgets continue to grow, the limited data volume provided by heavily filtered and deduplicated datasets will become a practical constraint. In efforts to better understand how to proceed, we study model performance at various compute budgets and across multiple pre-training datasets created through data filtering and deduplication. We find that, given appropriate modifications to the training recipe, repeating existing aggressively filtered datasets for up to ten epochs can outperform training on the ten times larger superset for a single epoch across multiple compute budget orders of magnitude. While this finding relies on repeating the dataset for many epochs, we also investigate repeats within these datasets at the document level. We find that not all documents within a dataset are equal, and we can create better datasets relative to a token budget by explicitly manipulating the counts of individual documents. We conclude by arguing that even as large language models scale, data filtering remains an important direction of research.

## 1 Introduction

Scaling data and compute has been the winning recipe for producing state of the art machine learning models. As costs increase with diminishing returns, recent work often points to high-quality data as a key factor for high model performance [30, 6]. One method of obtaining high quality data is through data filtering. However, as frontier level large language models (LLMs) scale in compute through increases in both parameter count and token count, there are concerns that we will run out of training data. Data filtering reduces the dataset size, creating a tension between data quality and data quantity. It is currently unclear whether the performance gains from data filtering can be sustained at scale through repetitions as an attempt to compensate for the reduction in dataset size.

Recent work has studied data filtering as a means of efficiently training strong models with limited compute [8, 15]. However, it is unclear if the findings are practical for training larger models. Unfortunately, existing frontier level LLMs lack transparency when it comes to their data filtering pipelines [30, 6, 1, 16]. This gap in knowledge adds to concerns about the practicality of data filtering on a scale.

Understanding the interaction between data filtering and scaling would be made easier if we better understood what it means to be a better dataset. Currently, the quality of a dataset is determined by using standardized benchmarks to evaluate the performance of a model trained on it. However, there are other traits we desire in the data beyond the performance of the raw downstream model, such as repeatability and performance over replacement. Developing a better understanding of these traits would allow for a more principled construction of better datasets.

39th Conference on Neural Information Processing Systems (NeurIPS 2025).

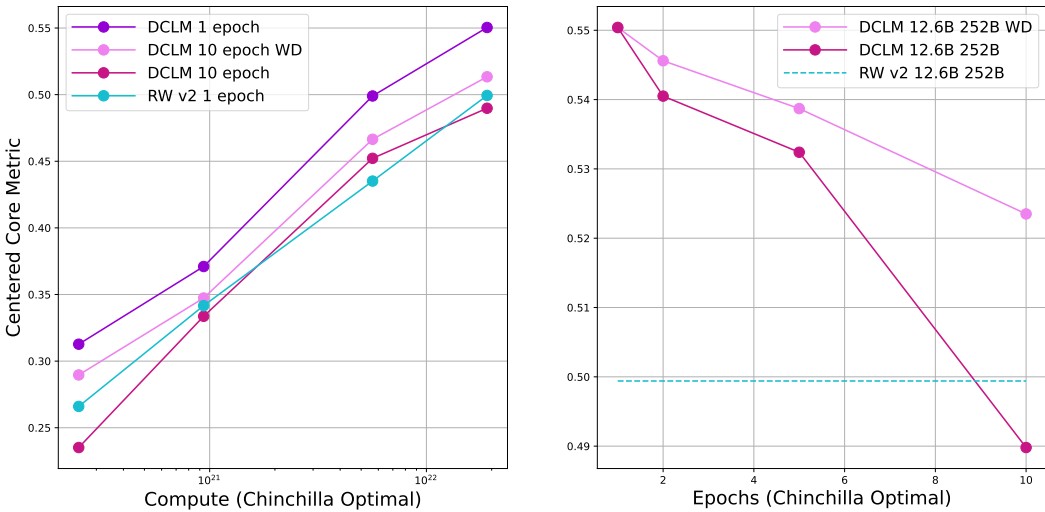

Figure 1: Repeating the aggressively filtered dataset DCLM-baseline for up to ten epochs consistently outperforms training on a single epoch of the ten times larger RefinedWeb supserset (cyan)—provided that we adapt the weight decay for the high-repetition runs. On the left, we show that this result appears consistent across compute budgets (model sizes of 1B, 3B, 7B, and 12.6B). On the right we show that, at the largest compute budget tested (12.6B parameters, 252B total tokens seen), adapting the weight decay as a function of repetition allows for a significantly better result versus training on the superset. Results are evaluated on the centered core metric from DCLM, which is a normalized average over 22 tasks. We also include a MMLU version of the right side in Appendix C.

In what follows, in Section 3 we study how datasets of different quality behave under repetition and explore how to improve multi-epoch performance. We compare DCLM-baseline filtering [15] with a superset comprising the strictly looser RefinedWeb filtering [21], as well as the commonly studied C4 dataset [23]. We find that while better datasets are not necessarily more repeatable, the interaction between dataset performance and repeatability depends on the compute allocation used to train the model. Furthermore, we study practical considerations for scaling up aggressively filtered datasets by repeating them for multiple epochs to account for the reduced number of tokens available. Datasets can be made more repeatable through regularizing factors, such as increasing weight decay and increasing tokens per parameter. We find that when training on the same number of total tokens, using ten epochs of the aggressively filtered DCLM-baseline can outperform a single epoch of its ten times larger superset—RefinedWeb.

Next, in Section 4 we investigate how to create better datasets by analyzing duplicate documents within the dataset. We find that web data filtering results in a bias towards documents that are more often partially duplicated, which was also reported in Penedo et al. [22]. Deduplicating the resulting dataset will therefore reduce the precision of the overall filtering step. We suggest simple methods that use a quality classifier to manipulate the individual counts of documents to improve dataset quality given a desired token budget, and show that this outperforms using only duplicate counts as in standard deduplication methods.

Lastly, we reflect on the role of data filtering for LLMs, and argue that it is a key ingredient for improving LLMs no matter the compute budget. We hope that this work not only clarifies some of the limitations of existing data filtering methods, but also demonstrates the value and practicality of data filtering, thus encouraging further research in this important direction.

## 2 Related work

**Data Filtering** We investigate whether data filtering is justifiable when it requires repeating to compensate for the decrease in tokens. Recent works such as DataComp-LM (DCLM) [15] and FineWeb-edu [22] have shown how data filtering can greatly improve performance. However, these datasets only retain a few trillion tokens, which may not be enough to train state-of-the-art models

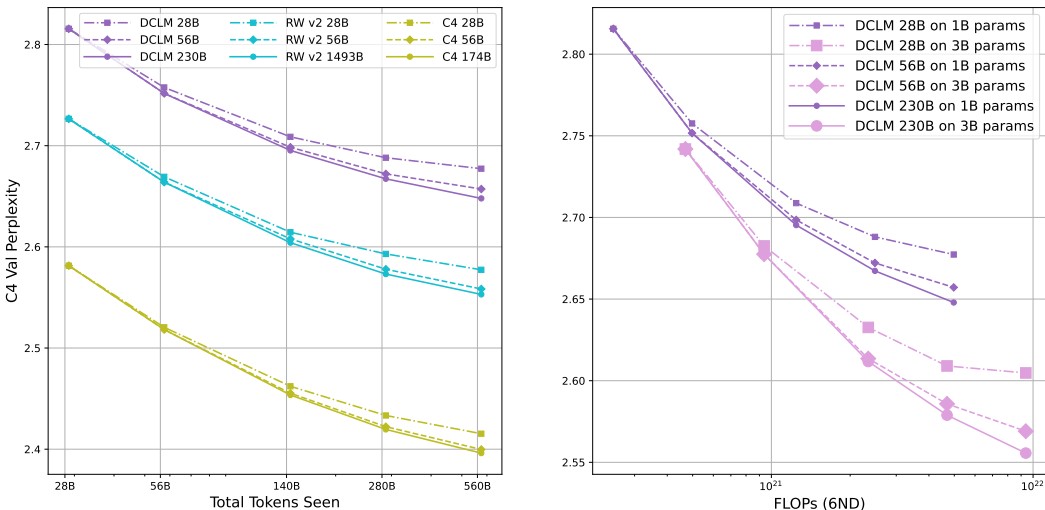

Figure 2: C4 perplexity as a function of training dataset and repetition count. Over-repeating data leads to diminishing returns irrespective of the training dataset chosen. For each dataset, we vary the number of unique tokens available, and then vary the training token budget. In the left plot we train 1B parameter models to show that heavy repetition results in a similar degradation in validation loss for all datasets tested. The right hand side shows that this effect holds at larger compute budgets for similarly over-trained models on the DCLM dataset.

unless trained for multiple epochs. Our study thus takes into account this scenario, where aggressively filtered datasets must be repeated, and compares them with larger but lower quality datasets.

**Repeatability**   We build on the work in Muennighoff et al. [19], which studies training models that are data-constrained. They find that for a fixed compute budget, web-crawl data can be repeated four times before encountering significant diminishing returns compared to training on unique tokens. We will refer to this as repeatability, which is measured for a fixed training token budget by comparing the performance of repeating a subset for multiple epochs with that of a single epoch. Unlike this work, we examine the repeatability of crawl-data from two new angles: 1. whether dataset quality affects repetition resilience; 2. if regularization techniques can dampen the impact of over-repetition.

Goyal et al. [9] studies data filtering for CLIP models and argues that data filtering methods must take into account the computational budget. They find that training a ViT-B/32 for 640M samples on 12.8M LAION filtered samples is worse than 128M unfiltered samples, and that CLIP models trained on large amounts of compute should use less aggressive filtering. Although this differs from our findings, there are significant experimental differences due to modality and filtering techniques.

**Compute Allocation**   Explicitly repeating tokens can lead to different optimal compute allocations. Hoffmann et al. [11] studies optimal allocation of parameters and tokens for a training computational budget. Their suggested ratio for parameters to tokens is called Chinchilla optimal, while training for more or less tokens can be considered overtraining or undertraining relative to that model size. Muennighoff et al. [19] find that for repeated data, overtraining is more efficient than staying at Chinchilla optimal.

**Deduplication**   Prior work has shown that training on deduplicated data is desirable because it improves performance [13], reduces privacy risks [12], and reduces memorization [3]. However, more recent works like DCLM and FineWeb-edu use sharded deduplication for engineering and performance reasons. TxT360 [17] takes a slightly different approach, using global deduplication followed by upsampling by the natural distribution. We build on existing deduplication methods, and move toward upsampling based on quality metrics on a document level.

# 3 Repeating Filtered Datasets

In parallel to LLMs training on larger datasets, researchers are also focusing on filtering data to improve quality. Together, these factors accelerate concerns about running out of training data. In the earlier days of training LLMs, common practice was to train LLMs on a single epoch of data. This is in stark contrast to state-of-the-art vision models, where practitioners would repeat data for many epochs. Though single epoch training may have benefits such as preventing memorization [13, 3], in this work we study data repetition across datasets from a purely performance perspective. Muennighoff et al. [19] studied the repeatability of datasets in a data-constrained setting and found that datasets can be repeated for four epochs before noticing diminishing returns compared to training on fresh data. Building off this, we examine datasets of varying quality created through data filtering to understand whether these data filtering approaches affect repeatability. In this section we study datasets "as is" and treat them as a single unit rather than a collection of documents. We use original C4, DCLM's reproduction of RefinedWeb (we will refer to this as RefinedWeb and RW v2 interchangeably), and DCLM-baseline. See Appendix I to compare datasets "as is" with datasets that are deduplicated.

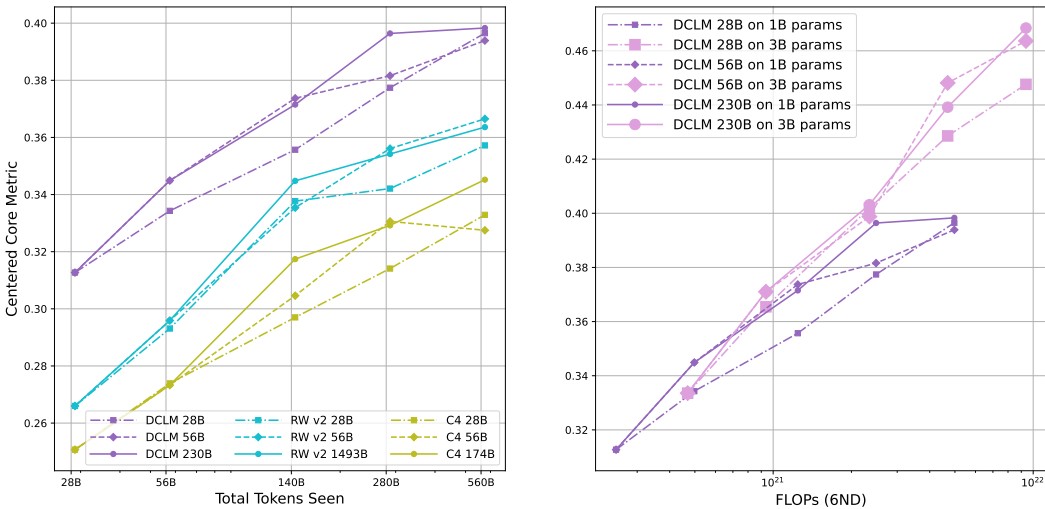

Figure 3: Downstream accuracy average of 22 tasks as different datasets are repeated and overtrained. The left side only contains 1B models across a variety of datasets and unique tokens, while the right side only trains on DCLM-baseline while varying in model size and unique tokens.

In Figure 2 we study a scenario similar to the findings in Muennighoff et al. [19], but instead start with Chinchilla optimal unique tokens for 1B parameter models. On the left side, we observe that the C4 validation perplexity eventually sees diminishing returns when training on C4 after repeating. In addition, we also train on higher quality datasets DCLM-baseline and RefinedWeb, and find that they have similar repeatability behavior when evaluating on C4 validation perplexity. In this setting we start with a Chinchilla optimal compute allocation and then repeat epochs by fixing the model size while increasing the token count, and compare this to datasets with more available tokens. On the right side we vary model size, but maintain the same dataset sizes that we repeat. This figure suggests that increasing model size while maintaining dataset size decreases repeatability, but there is not enough information to make a conclusion so we will explore this further later in the section. When we move to evaluating downstream accuracy across a 22 task subset from LLM Foundry (referred to as core centered metric in DCLM), we see in Figure 3 that quantifying repeatability is even less clear. It appears that when comparing using downstream accuracy, repeatability is more difficult to measure, and it is unclear if repeating meets the same diminishing returns seen when evaluating C4 perplexity. What is common between the two evaluation settings is that datasets that achieve better performance do not appear to be more repeatable than inferior datasets when examining diminishing returns.

Although better performing datasets are not more repeatable by preventing diminishing returns, in the overtraining setting (Figure 3) better datasets repeated multiple times clearly maintain better performance than an inferior dataset that is not repeated. However, it is important to note that

overtraining is practical only in certain settings with priorities such as minimizing the inference cost, and there is an assumption in Figure 3 that there are at least enough tokens to train Chinchilla optimal. This assumption does not always hold when LLMs are scaled up, especially when data filtering is involved.

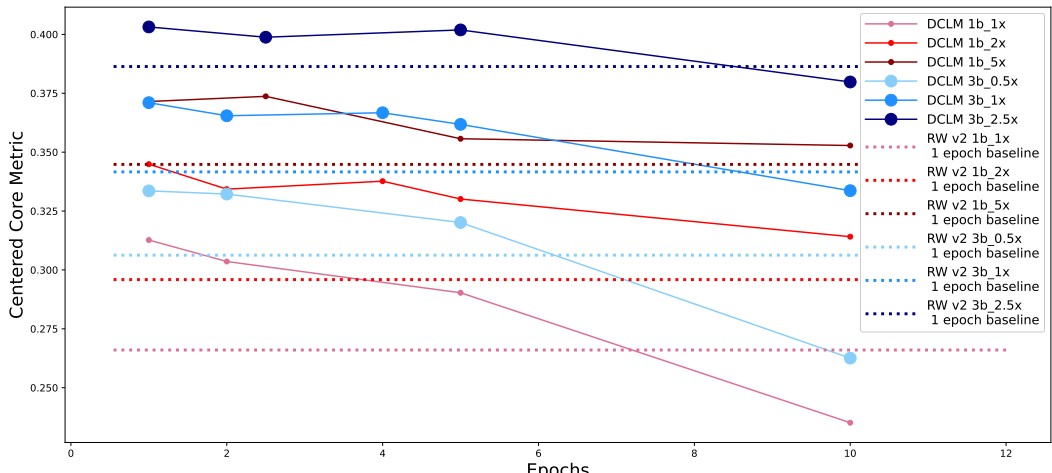

Figure 4: Downstream accuracy average of 22 tasks of DCLM-baseline dataset as we fix total tokens trained and vary epochs and unique tokens seen. Dotted horizontal line represents training one epoch on the RefinedWeb baseline, which we compare against because DCLM-baseline is additional filtering on top of RefinedWeb. Legend denotes model size and multiplier to scale tokens seen at Chinchilla optimal.

In contrast to repeating datasets by training for longer, we next examine fixing the token budget and repeating by decreasing the number of unique tokens. This scenario forces us to examine cases where there are fewer unique tokens available than Chinchilla optimal. In Figure 4 we compare for a fixed model size and total token budget how performance varies as you reduce the number of unique tokens available, causing an increase in the number of repeats for which the model sees the dataset. When there is no longer at least the Chinchilla optimal number of tokens available, repeating the DCLM-baseline dataset with too few unique tokens can lead to worse performance than training for a single epoch on its superset of RefinedWeb filtered data. On the other hand, compute allocations with high tokens per parameter seem to benefit more from higher quality datasets, since DCLM can be repeated more times to equalize performance with unrepeated RefinedWeb. Tokens per parameter is the main factor, as increasing model size while maintaining token count decreases DCLM repeatability to equalize with RefinedWeb, and increasing token count while maintaining tokens per parameter does not consistently increase DCLM repeatability to equalize performance with unrepeated RefinedWeb (Figure1, left). This would also explain why unrepeated RefinedWeb does worse than heavily repeated DCLM in Figure3. But there are increasing diminishing returns to performance with respect to compute when increasing tokens per parameter, so there still needs to be a balance when doing data filtering.

Table 1: Weight decay can reduce performance degradation from repeating data. All models are 12.6B parameters trained for 252B total tokens. We report results using the centered core metric. The default weight decay is 0.0316. Note that we confirm WD 1x is optimal for a single epoch of RefinedWeb, as well as for a single epoch of DCLM.

| Repeats | WD 1x | WD 2x | WD 3x |
|---|---|---|---|
| 10 epochs | 0.489 | 0.523 | 0.513 |
| 5 epochs | 0.532 | 0.539 | 0.528 |
| 2 epochs | 0.541 | 0.546 | 0.545 |
| 1 epochs | 0.550 | 0.547 | 0.546 |

We can mitigate performance degradation caused by repeating data with regularization in the form of increasing weight decay. In Table 1, we see that the optimal weight decay value increases as we increase the number of repeats. A reasonable schedule is to scale weight decay by roughly the square root of the number of repeats. With the optimal weight decay values, we see in Figure 1 that training for ten epochs of DCLM-baseline outperforms training on RefinedWeb filtered data for a single epoch. This is notable because DCLM-baseline takes the top 10% of documents of RefinedWeb as scored by a FastText classifier. However, it is important to note that this does not mean DCLM-baseline is a strictly better dataset than RefinedWeb filtering. When data is a much more constrained resource than compute, it is likely that diminishing returns from heavily repeating DCLM-baseline will perform worse than repeating RefinedWeb filtered data a couple of times.

## 4 Repeating Documents To Create Better Datasets

Table 2: 7B models trained on DCLM-baseline for 138B tokens (Chinchilla optimal) using different subsampling methods.

| Subsampling Method | CORE (22 task average) | MMLU |
|---|---|---|
| Duplicate-aware subsample | 34.1 | 25.1 |
| Global deduplication then subsample | 41.7 | 28.5 |
| Uniform subsample | 44.4 | 42.2 |
| Keep docs with duplicates $\geq 7$, then dedup | 45.8 | 40.0 |

We have established that, even as compute budgets increase, models can remain robust to dataset repetition given the right training recipe. We now investigate whether high quality data repetition may be best implemented at a document level, as opposed to increased epochs on the entire dataset. To better understand this question, we dive into DCLM-baseline, which contains many duplicates (fuzzy and exact) because it uses sharded deduplication due to engineering constraints. Penedo et al. [22] also uses sharded deduplication, observing that global deduplication does not always improve performance. This is in contrast to prior standard practices of deduplication, which recommended global deduplication and training for a single epoch on the resulting dataset.

The performance gains from data filtering clearly demonstrate that not all documents are of equal quality. We can also observe this by comparing subsampling strategies. The standard strategy is uniform subsampling, which results in a dataset that is diverse with very few duplicates, while more likely keeping documents that have more duplicates. A second strategy is to run minhash-based global fuzzy deduplication before subsampling down to the desired token count. Compared to uniform subsampling, incorporating global fuzzy deduplication treats documents with different duplicate counts equally, but both strategies should have few or no duplicates in the resulting subset if subsampling significantly. Lastly, we can mimic the duplication profile of the original dataset by using duplicate-aware subsampling. For each group of duplicates, this strategy keeps or removes the entire group with probability equivalent to the duplicate count normalized by the desired dataset reduction. This strategy results in a subset with a significant number of duplicates compared to the other subsampling strategies.

In Table 2 we observe that duplicate-aware subsampling performs the worst, suggesting that keeping a significant number of duplicates hurts performance. Uniform subsampling does significantly better than global deduplication before subsampling, which shows that documents in DCLM-baseline with high duplicate count are likely to be of higher quality. This is further supported by the fact that keeping unique copies for documents with greater than seven duplicates achieves competitive performance to uniform; however, this removes over 96% of the documents in DCLM-baseline.

Given a token budget and a metric correlated with document quality, we can take advantage of these findings by manipulating counts of individual documents through resampling. This is analogous to mixing but at document level instead of forming buckets. We use a function for assigning the counts of deduplicated documents within the dataset, adjust the range of the function by estimating how many more times you should repeat the best document in the dataset before including the worst, and adjust the domain of the function based on the number of documents desired. Figure 5 contains examples of such functions, which include greedily selecting the best documents with a constant function equal to a certain count (y=count), and a linear function which would increase the diversity

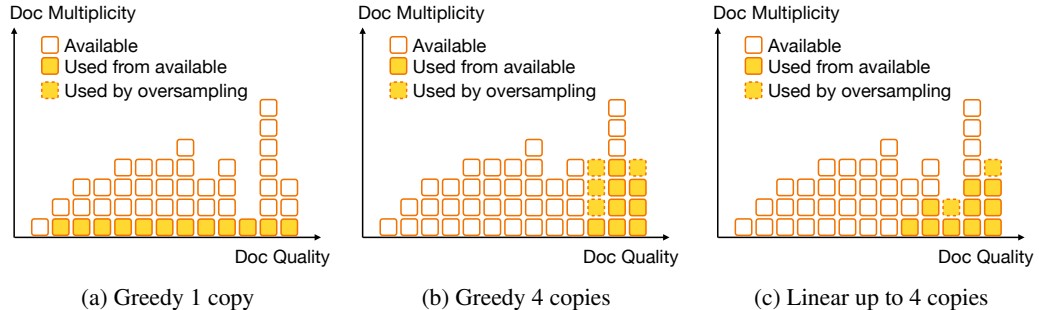

| (a) Greedy 1 copy | (b) Greedy 4 copies | (c) Linear up to 4 copies |

Figure 5: Examples of strategies for count manipulation. Note that Greedy 1 copy is similar to global deduplication, and can result in worse performance if document quality varies significantly. For example, if the dataset is split evenly between high quality documents and low quality documents, it may be better to repeat the high quality documents.

Listing 1: Pseudocode to sample documents using count manipulation with a simple greedy function.

```python
def greedy_function_example(score):
    # given a score, return how many copies of the accopanying
    # document we want.
    # threshold is pre-calculated from all scores.
    # in this example, threshold keeps 1 copy of top
    # documents to reach desired output document count.
    if score >= threshold:
        return 1
    else:
        return 0

def sample_count_manipulation(documents, counts, scores, function):
    output = []
    for doc, count, score in zip(documents, counts, score):
        target_count = function(score)
        for _ in range(target_count):
            if np.random.rand() <= 1 / count:
                output.append(doc)
    return output
```

of documents when compared to the equivalent greedy strategy with the same maximum count value. Key to this approach is having a metric correlated with document quality. Here we use the worst ranking within the dataset between FastText score and pre-deduplication document count. In Table 3 we subsample from the 3.8T DCLM-baseline down to 138B tokens for training a 7B model, and our results show that doing basic document count manipulation can outperform the various baseline subsampling approaches. In this setting of extreme subsampling that reduces trillions of tokens to billions of tokens, the best strategy is greedily selecting high quality documents according to our metric without repeating.

Table 3: 7B models trained for 138B tokens on DCLM-baseline with different document count manipulation strategies. Figure 5 visualizes these strategies. Ensemble uses the worst rank between FastText score and duplication count.

| Subsampling Method | CORE (22 task average) | MMLU |
|---|---|---|
| DCLM-baseline uniform subsample | 44.4 | 42.2 |
| Greedy 1 copy (dup count) | 41.7 | 28.5 |
| Greedy 4 copies (ensemble) | 44.6 | 38.5 |
| Greedy 1 copy (ensemble) | 46.1 | 46.0 |
| Linear up to 4 copies (ensemble) | 46.0 | 44.2 |

Table 4: 7B models trained for 138B tokens by doing count manipulation on a duplicate-aware subsampled 280B tokens from DCLM-baseline. In this setting, the greedy yet diverse strategy sees each document approximately 3 times. Metric used is FastText score. Note that using duplication count as the metric would be ineffective here because there are not enough unique documents.

| Subsampling Method | CORE (22 task average) | MMLU |
|---|---|---|
| DCLM-baseline DA subsample | 34.1 | 25.1 |
| Greedy Diversity | 40.4 | 28.3 |
| Greedy 5 copies | 40.4 | 29.4 |
| Linear up to 5 copies | 41.2 | 30.4 |

The results in Table 3 are unusual in that we are taking an already highly filtered dataset and subsampling even further. Naturally, there are enough high quality tokens to train without repeats, hence the success of the greedy yet diverse strategy. However, this becomes impractical as we scale and there are no longer enough unique tokens to support this strategy. Therefore we now investigate the setting where the dataset we are trying to manipulate is closer in size to the desired token count. We duplicate-aware subsample DCLM-baseline down to 280B tokens and apply count manipulation to produce 138B tokens for training a 7B model. As we scale back up, this would allow us to use the strategy on the entire DCLM-baseline to create a dataset close to two trillion tokens. In Table 4, we see that the best strategy is applying a linear function to balance the tradeoff between exploring diverse documents and exploiting high quality documents. Contrasting the best strategy in the two settings suggests that using a good count manipulation strategy is more important in the setting where data is constrained.

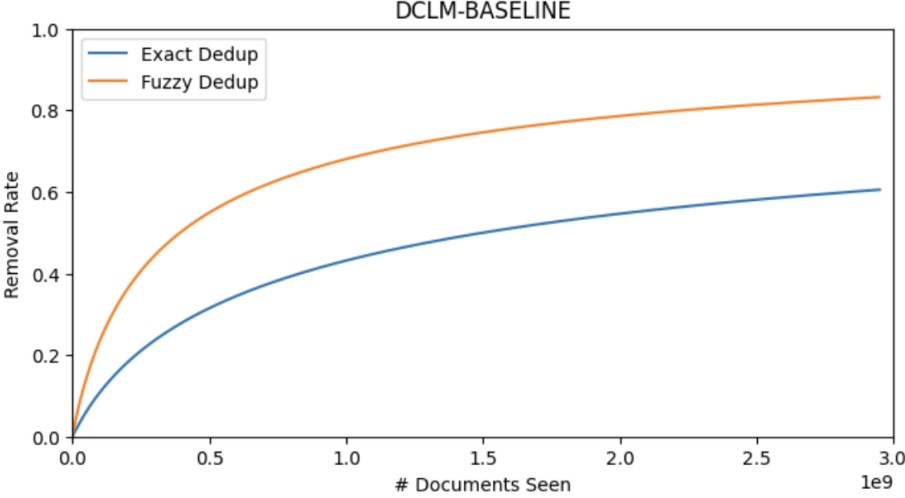

Figure 6: In DCLM-baseline, the proportion of duplicates inside the dataset increases as the number of documents increases. In Appendix E we see this trend also holds for RefinedWeb, but with lower removal rates.

Combining duplicate-aware subsampling with count manipulation is a powerful tool for studying datasets. Increasing dataset size typically refers to obtaining new tokens of the same quality. DCLM does this by increasing the number of Common Crawl WARC files at the same rate as increasing the training token budget. However, as seen in Figure 6, when the number of WARC files increases, so does the number of duplicates. We find that DCLM-baseline constructed from the DCLM 7B_2x pool contains 33% fuzzy duplicates, but DCLM-baseline constructed from 20 times more WARC files contains 83% fuzzy duplicates. By uniformly subsampling to run smaller scale experiments, this experiment design uses both new data and repetitions when scaling back up. Additionally, it is impractical to deduplicate unfiltered data due to engineering constraints caused by the amount of data, as well as design decisions like what type of deduplication to use. Duplicate-aware subsampling gives us the tools to design experiments where data scaling happens through providing new data of equal quality.

Count manipulation differs from filtering because filtering is restricted by the amount of documents available in the pool. Even with the perfect data quality metric, filtering alone may not create the best dataset because high quality documents do not necessarily appear many times in the dataset. Count manipulation allows for oversampling and is thus complementary to filtering, depending on traditional filtering techniques like deduplication and filtering based on a quality metric. Furthermore, while the experiments presented have built off of DCLM-baseline, it is important to remember that DCLM-baseline is an aggressively filtered dataset. Count manipulation should be just as if not more impactful on less filtered datasets because there is a wider range in quality of documents within the dataset, making it more likely that repetition of high quality documents improves performance more than seeing low quality documents for the first time.

Revisiting the ideas in Section 3, we can replace repeating entire datasets with count manipulation. Though it may appear that datasets as a whole can only repeat a set number of times before significant diminishing returns, when examining individual documents it is likely that repeating a high quality document many times is still better than seeing a low quality document for the first time. This tradeoff can be captured by the function applied to count manipulation.

# 5 How to Use Better Datasets

We would like to conclude by arguing that creating better datasets is still an important area of research. Our findings in Section 3 suggest that an aggressively filtered dataset like DCLM-baseline can be used as pre-training data at scale by repeating the dataset for multiple epochs with the right recipe adaptations, but this is just one of the many use cases for data filtering. Additionally, our findings in Section 4 can be used to improve existing datasets, and can be especially useful in situations where there is a constraint on the availability of unique data.

Better datasets are especially beneficial for training smaller scale LLMs. This can happen when practitioners want to reduce inference cost by training a smaller model or when there is a limit on the amount of compute available. In this setting, the amount of data available is no longer a key constraint, which places a greater importance on the tokens that are actually seen. Recent works such as DCLM [15] and FineWeb-edu [22] have shown how data filtering can greatly improve performance across models that only vary by the training data used.

High quality data is often used during continual pre-training. Works like DCLM and Llama 3 [6] use this technique to introduce higher quality data at the end of pre-training. This technique is separate and often complementary to fine-tuning techniques like instruction tuning. However, we find in Appendix D that for 7B models trained on 138B tokens, using DCLM-baseline for the last 10% or 20% of training performs the same as mixing that same proportion into RefinedWeb. Nonetheless, this technique can be used to introduce high quality data that was not ready at the beginning of the training run, and may help more for smaller amounts of data or different domains. Feng et al. [7] have shown that curriculum learning can be effective when targeting particular domains like math and code.

Lastly, while DCLM-baseline may not contain sufficient tokens for all use cases, it is important to remember that almost all pre-training datasets have a filtered component and we can change the filtering ratio based on needs. Throughout this work we have compared DCLM to RefinedWeb because DCLM is created through additional filtering on top of RefinedWeb filtering. As compute budgets for LLM pre-training increase, more emphasis will be placed on looser and more efficient data filtering that achieves high recall.

**Limitations** Data filtering techniques have their limitations as well. Our understanding of models is only as good as the evaluations we rely on. It is possible that data filtering reduces capabilities or removes knowledge that we desire in our models, but we can only hope to quantify this through evaluations that measure the effects that we care about. Additionally, we focus strictly on web-scraped datasets created for general language modeling. Dataset properties may differ when studying different domains such as code and math. Finally, our experiments are smaller scale than frontier level models, so our trends must continue to hold as we scale in order for these findings to apply to frontier level models.

# Impact Statement

This paper studies data filtering and its role in improving Large Language Models. The societal consequences are similar to those typically discussed in Machine Learning and Large Language Models. One additional consequence we would like to highlight is the potential for bias amplification in dataset creation. Although we are not releasing additional datasets and models, repeating or resampling data can contribute toward certain biases, so researchers should take proper care to evaluate for such when developing and releasing datasets and models.

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

## A  Training Setup

We use OpenLM (MIT license) for training. Standard model configurations and hyperparameters can be found in the DCLM code base. OpenLM is used for all 1B and 3B models, as well as 7B models in Section 4. These models are trained on Nvidia H100s.

We use AXLearn for 7B and 12B models in Section 3. These models are trained on TPUs.

The main datasets we trained on are DCLM (CC-by-4.0), RefinedWeb (v2, from DCLM) (CC-by-4.0), and C4 (CC-by-4.0 or odc-by depending on source).

## B  Evaluation

We evaluate on the centered core metric from DCLM, which consists of the following 22 tasks: AGI Eval LSAT Analytical Reasoning [33], ARC Challenge [5], ARC Easy [5], BIG-bench cs algorithms [28], BIG-bench dyck languages [28], BIG bench language identification [28], BIG-bench operators [28], BIG-bench: wikidata [28], BIG-bench repeat copy logic [28], BoolQ [4], Commonsense QA [29], COPA [26], CoQA [25], HellaSwag (zero-shot and few-shot) [32], Jeopardy (MosaicML), LAMBADA [20], OpenBook QA [18], PIQA [2], SQuAD [24], Winograd Schema Challenge [14], Winogrande [27]

We also evaluate 7B and larger models on MMLU (few-shot) [10].

## C  MMLU Repeatability

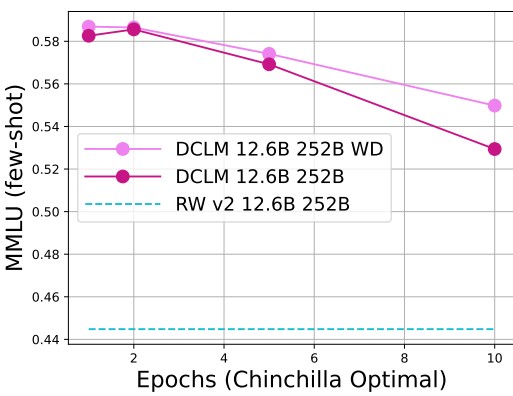

Figure 7: Right side of Figure 1 except measuring MMLU (few-shot) on the y-axis. We see that DCLM even with repeats significantly outperforms RefinedWeb on MMLU, even more so when comparing against centered core metric.

## D  Continual Pre-training

We train 7B models for 138B tokens to investigate the effect of curriculum learning on general pre-training data of different quality.

| Dataset | Core | MMLU | Extended |
|---|---|---|---|
| Shuffle 90% RW and 10% DCLM | 41.3 | 25.7 | 22.4 |
| 90% RW followed by 10% DCLM | 41.0 | 26.2 | 21.9 |
| Shuffle 80% RW and 20% DCLM | 41.6 | 27.3 | 22.5 |
| 80% RW followed by 20% DCLM | 41.6 | 26.1 | 22.2 |

# E   Deduplication Trends as Datasets Scale

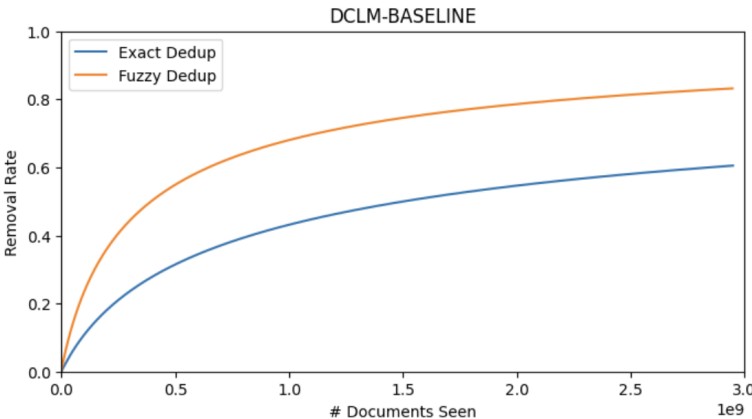

Figure 8: In DCLM-baseline, the proportion of duplicates inside the dataset increases as the number of documents increases. In Figure 9 we see this trend also holds for RefinedWeb, but with lower removal rates.

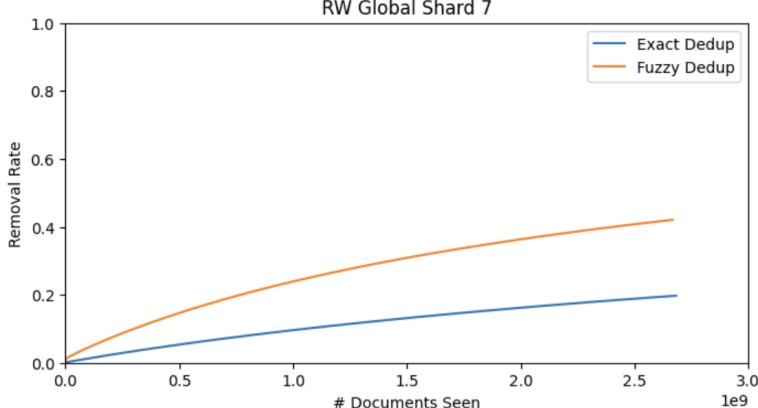

Figure 9: In RefinedWeb, the proportion of duplicates inside the dataset increases as the number of documents increases. Notice that these removal rates are lower than those of DCLM-baseline.

# F  Fuzzy vs Exact Duplicates

Some initial 7B parameter (Chinchilla optimal) experiments to explore the differences between fuzzy and exact duplicates. For the second pair of experiments, floor f refers to keeping documents that have at least f fuzzy copies, while ceil c refers to keeping no more than c copies. Additional work is required to better understand the differences between fuzzy and exact duplicates.

| Dataset | Core | MMLU | Extended |
|---|---|---|---|
| DCLM-baseline duplicate-aware subsample (fuzzy) | 34.1 | 25.1 | 17.5 |
| DCLM-baseline duplicate-aware subsample (exact) | 33.8 | 25.3 | 17.7 |
| DCLM-baseline floor 21 ceil 4 (fuzzy) | 42.7 | 37.4 | 25.6 |
| DCLM-baseline floor 21 ceil 1, 4 epochs (exact) | 43.3 | 32.4 | 24.0 |

# G  DCLM-baseline Count Manipulation Metric Statistics

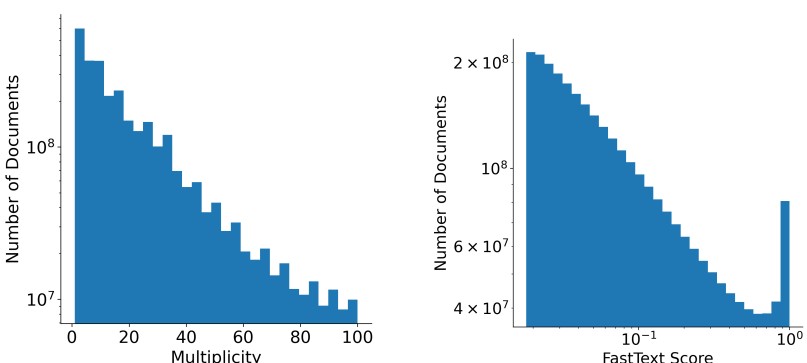

(a) DCLM-baseline distribution of fuzzy duplicates

(b) DCLM-baseline distribution of FastText scores

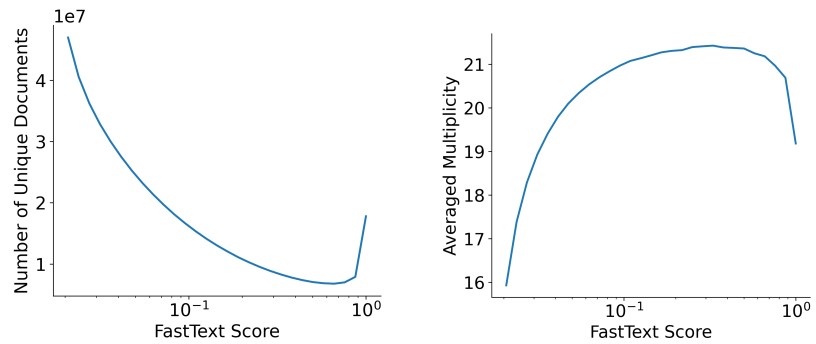

(c) DCLM-baseline unique documents by FastText score

(d) DCLM-baseline average fuzzy duplicate count by FastText score

# H   Deduplication Methods Details and Comparisons

We use minhash-based fuzzy deduplication written in Rust. Our hyperparameters are 5-gram tokens and 14 bands of size 9.

Here we compare our fuzzy dedup results against the implementation from Tokpanov et al. [31] (Zyda-2 DCLM-dedup), as well as DCLM-baseline (no additional global dedup).

Table 5: 7B Chinchilla optimal deduplication comparison

| Dataset | Core | MMLU |
|---|---|---|
| DCLM Our Fuzzy Dedup | 41.7 | 28.5 |
| Zyda-2 DCLM-dedup | 43.6 | 29.4 |
| DCLM-baseline | 44.4 | 42.2 |

Table 6: 12B Chinchilla optimal deduplication comparison

| Dataset | Core | MMLU |
|---|---|---|
| Zyda-2 DCLM-dedup | 53.3 | 56.8 |
| DCLM-baseline | 55.0 | 58.3 |

# I   Deduplication Interaction with Repeats

In Section 3, we study datasets as is, leaving the possibility of duplicates already existing in the datasets. Here we compare repeating a randomly sampled subset with repeating a global fuzzy deduplicated (Zyda-2 DCLM-dedup) subset. Note that randomly subsampling produces datasets where documents either have no or low repeats because the subset is much smaller than the pool that we subsample from, and the datasets studied in Section 3 already go through some sort of deduplication.

The results below suggest that low repeat counts favor DCLM-baseline, while high repeat counts favor the deduplicated version. At the compute regimes we study our findings should apply to both scenarios, but we hypothesize that deduplication or resampling becomes more important as we use higher repeat counts at larger scales.

| Epochs | DCLM Core | Dedup Core | DCLM MMLU | Dedup MMLU |
|---|---|---|---|---|
| 1 epoch | 55.0 | 53.3 | 58.3 | 56.8 |
| 2 epoch | 54.0 | 53.5 | 58.6 | 57.9 |
| 5 epoch | 53.2 | 51.9 | 56.9 | 55.5 |
| 10 epoch | 49.0 | 49.5 | 52.9 | 51.4 |

# J   Additional Count Manipulation Details

## J.1   Count Manipulation Strategies

One way to determine the count function corresponding to each of the strategies in Figure 5 is as follows:

1. Greedy 1 copy: Find the threshold such that the desired number of documents after deduplication is above the threshold. Then sample all documents probabilistically by keeping each document with probability $\frac{1}{duplicate\_count}$ if the document's score is over the threshold.

2. Greedy 4 copies: Find the threshold such that the desired number of documents after deduplication is above the threshold. Then sample all documents probabilistically by looping 4 times per document over the threshold, with each trial attempting to keep that document with probability $\frac{1}{duplicate\_count}$.

3. Linear up to 4 copies: First, determine the maximum number of copies (in this case $max\_copies = 4$) you would like to keep for the best document. Next, determine the goal number of document ($goal\_docs$) you would like to keep. The number of unique documents for each copy count is $bucket\_size = \frac{goal\_docs}{\sum_{i=1}^{max\_copies} i}$. Using this, calculate the top $max\_copies$ thresholds using the deduplicated documents. Lastly, for each document (non-deduplicated), use the thresholds to identify which copy count bucket ($i$) it belongs to, and loop $i$ times, with each trial attempting to keep that document with probability $\frac{1}{duplicate\_count}$.

## J.2   Ensembling

The ensembling strategy in Table 3 is calculated by first finding each document's duplication count and DCLM FastText classifier score, then assigning each document its ranking (e.g. best score is 1, worst score is number of documents) for each metric, and finally taking the maximum of those values. In this new metric, lower is now better. It is important for each metric used in the ensemble to not only be correlated with data quality, but also be fine-grained. Otherwise, in Table 4 duplication count is no longer a useful metric because there are not enough unique documents.

## J.3   Metric Comprison Between FastText Score and Ensemble

Table 7: Ensemble results from Table 3 compared with using DCLM FastText score as metric.

| Subsampling Method | CORE (22 task average) | MMLU |
|---|---|---|
| DCLM-baseline uniform subsample | 44.4 | 42.2 |
| Greedy 1 copy (dup count) | 41.7 | 28.5 |
| Greedy 4 copies (ensemble) | 44.6 | 38.5 |
| Greedy 1 copy (ensemble) | 46.1 | 46.0 |
| Linear up to 4 copies (ensemble) | 46.0 | 44.2 |
| Greedy 4 copies (FastText) | 41.6 | 45.3 |
| Greedy 1 copy (FastText) | 45.5 | 45.4 |
| Linear up to 4 copies (FastText) | 42.1 | 46.7 |

