# OpenReview forum: "Datasets, Documents, and Repetitions: The Practicalities of Unequal Data Quality"
_NeurIPS.cc/2025/Conference — NeurIPS 2025 poster_

### Official Review · Reviewer_Ta7J · 2025-07-05

**Clarity:** 2
**Significance:** 3
**Originality:** 3
**Rating:** 4
**Confidence:** 4

**Summary:**

This paper investigates how to duplicate data samples under many different practical data settings. This is a timely topic since as LLMs scale, the compute need for data, as the chinchilla optimal rule suggested, also grows. The paper examines model performance across various compute budgets and pre-training datasets that have undergone filtering and deduplication.

The paper provides several findings, mostly are intuitive and are aligned with community beliefs. For example, higher quality datasets may be affordable to be repeated more times, and that not all documents are equal, and better datasets can be created by manipulating individual document counts considering document quality.

**Questions:**

- Given the paper's crucial observation that adjusting weight decay is key for effective data repetition, I am curious to understand the authors view if the findings align with, or diverge from, the theoretical insights presented in papers like [1].


[1] https://arxiv.org/abs/2502.15938

**Ethical Concerns:**

["NO or VERY MINOR ethics concerns only"]

**Final Justification:**

The writing of this paper is probably one of the biggest concern, as pointed out by two reviewers.

While I am writing this final justification, I am still waiting the authors to provide more updates. As the state of this, even though the authors agree to update the paper. There are significant rewriting needed and I am not convinced that we will get a manuscript up to the standard of the venue. I am not leaning towards acceptance, and may even consider lower the scores.

**Limitations:**

Yes

**Quality:**

3

**Strengths And Weaknesses:**

# Strengths:

## Practical considerations and scenarios

It is commendable the authors for thoroughly exploring a diverse range of distinct and practical data constraints and scenarios. Unlike many dataset papers that make a particular assumption of constraint and conduct experiments solely around that, which can sometimes limit practical applicability, this paper investigates scenarios such as:

- Reproducing Muennighoff et al.'s work on diminishing return of repated data
- Simulating scenarios where "the dataset we are trying to manipulate is closer in size to the desired token count"
- "fix the token budget and the number of unique tokens used for training"

This broad practical scope significantly enhances the utility and relevance of the findings for real-world LLM training. These results will be valuable to different training goals, such as small models, large models, inference-optimal models, etc.

## Practical findings and training methods

The paper finds that adjusting weight decay will help reduce the diminishing return of repetition, and also provides a practical data sampling method. The weight decay method remind me of [1], where the authors theoretically study the interplay of weight decay and data set size, by considering the model weights as a EMA of the weight updates. This finding here highlights that dataset study should not be considered simply as a "preprocessing step" before training.

The study offers actionable strategies for optimizing LLM training with constrained resources, whether due to limited unique data volume or specific compute budgets.

# Weakness

## Writing
The paper presents a list of distinct experimental settings and comparisons. However, the overall organization and presentation could be improved to make it easier for readers to find and consistently follow the flow of these numerous experiments, sometimes lacking clear initial roadmaps or explicit transitions between different scenarios.


[1] https://arxiv.org/abs/2502.15938

---

> ### Author Rebuttal · Authors · 2025-07-31
>
> Thank you for your helpful suggestions.
>
> Although PDF uploading is disabled, we have improved our draft by adding an explicit roadmap at the end of the introduction, as well as additional transitions and clear lead sentences for paragraphs introducing new experiments.
>
> Regarding your question, figure 4 of Bergsma et al. suggests that weight decay is more important as you increase tokens per parameter, but their data is deduplicated and is not repeated. Our paper explores weight decay in relation to data repetition, so perhaps these findings are orthogonal. One potential contradictory point is that with repetition in mind, weight decay does matter at chinchilla optimal, which they would consider in the lower TPP regime.

---

> > ### Comment · Reviewer_Ta7J · 2025-08-03
> >
> > Thanks for acknowledging the structure update.
> >
> > As for the question, In the paper I mentioned, the idea is that model update can be considered as a EMA of the weight updates, I am curious too see if this provides a theoretical justification of why one need to adjust weight decay when there are more duplicates. I am providing this perspective only as a question for research discussion, hoping to see if there are some theoretical justifications. If the authors find this to be orthogonal or out-of-scope, free feel to ignore this question for now.

---

> > > ### Comment · Reviewer_Ta7J · 2025-08-06
> > >
> > > The writing of this paper is probably one of the biggest concern, as pointed out by two reviewers.
> > >
> > > While I appreciate the authors try to improve the structure. I probably need more evidence to show that the revised manuscript is up to the standard of NeurIPS.

---

> > > > ### Author Response · Authors · 2025-08-07
> > > >
> > > > Thank you for your comments. We will continue to iterate on the structure and writing to improve clarity and presentation. Other than the aforementioned changes, were there additional things you believe would improve the writing quality?

---

> > > > > ### Comment · Reviewer_Ta7J · 2025-08-08
> > > > >
> > > > > Thanks. Thew aforementioned changes are good. I am just not sure how I can confidently recommend the paper for publication without seeing the actual manuscript. I guess it is not really possible. So far I am still keeping my score as "borderline accept"

---

### Official Review · Reviewer_w2eL · 2025-07-05

**Clarity:** 2
**Significance:** 3
**Originality:** 4
**Rating:** 5
**Confidence:** 4

**Summary:**

This paper tackles the tension between data quality and data quantity in training large language models under a fixed compute budget. The authors show that, once you’ve exhausted “fresh” high-quality data (DCLM), you can squeeze extra performance by repeating that subset—provided you adjust regularization (e.g., increase weight decay). They then push this further with a document-level count manipulation scheme: assigning repeat counts to each doc based on a quality score, so that the constraint of compute budget focuses only on better corpus.

**Questions:**

1. Could you provide a more detailed sensitivity analysis of weight decay across repeat counts?

2. You use FastText-based scores to rank documents. Have you tried other proxies (e.g., language-model perplexity, diversity metrics, QuRater)? Does the choice materially affect outcomes?

3. Document-level count manipulation introduces bookkeeping overhead. Can you quantify the additional preprocessing time and memory compared to uniform sampling?

4. A single “pipeline” figure showing: (a) scoring, (b) count assignment function, (c) repeat schedule, would greatly improve clarity. Consider adding this to the camera-ready.

**Ethical Concerns:**

["NO or VERY MINOR ethics concerns only"]

**Final Justification:**

The authors have replied to my question and address my major concern. I have raise my score accordingly.

**Limitations:**

yes

**Quality:**

3

**Strengths And Weaknesses:**

Strengths:
- The paper spans multiple model sizes, a wide range of tasks, and detailed ablations (perplexity, core metrics, MMLU), which makes the findings very robust.

- Beyond just “here’s a trick,” the authors provide hyperparameter settings, pseudo-code for count manipulation, and appendices with implementation details—bonus points for reproducibility.

- While prior work has touched on data filtering and sampling, the explicit pairing of multi-epoch repeats, plus the doc-level repeat allocation, limited-resource budget regimes.


Weaknesses:

- Key insights (e.g., that boosting weight decay is required to make repeats helpful) are buried. Figure 1’s default curves actually *mislead* unless you already know to jump to Table 1. There’s no high-level roadmap or a guiding “flow” diagram, so it reads like disjointed blog posts stitched together.

- The “why” behind count manipulation isn’t motivated with simple analogies or visuals. Readers unfamiliar with corpus building jargon may struggle to grasp why document-level repeats beat global strategies.

- Juggling four different model scales × three evaluation metrics × multiple sampling schemes makes it hard to see the forest for the trees. A more focused presentation (or a distilled summary table) would help.

---

> ### Author Rebuttal · Authors · 2025-07-31
>
> Thank you for your helpful comments towards improving our work.
>
> We have improved our draft by adding an explicit roadmap at the end of the introduction, as well as additional transitions and clear lead sentences for paragraphs introducing new experiments. Additionally, we motivate count manipulation by connecting Figure 5a with standard deduplication, and expanding the caption.
>
> New Figure 5 caption: Examples of strategies for count manipulation. Note that Greedy 1 copy is similar to global deduplication, and can result in worse performance if document quality varies significantly. For example, if the dataset is split evenly between high quality documents and low quality documents, it may be better to repeat the high quality documents rather than each document a single time.
>
> For figure 1, would adding an arrow between the purple and pink lines labeled “improvement from weight decay” improve clarity?
>
> Regarding your questions:
>
> 1. Table 1 contains results where we vary weight decay and repeat counts. Here is also an additional table that contains the same but for 3B models trained for chinchilla optimal number of tokens:
> | Repeats | WD 0.033 | WD 0.06 | WD 0.1 |
> | ----- | ----- | ----- | ----- |
> | 10 epochs | 0.334 | 0.332 | 0.347 |
> | 5 epochs |  ​​0.362 | 0.367 | 0.367 |
> | 2 epochs | 0.365 | 0.372 | 0.366 |
> | 1 epoch | 0.371 | 0.373 | 0.372 |
>
> 2. The main scores we used were FastText and duplication count, and then ensembling the two. We find that FastText performs slightly better than duplication count alone, and ensembling the two provides some additional improvement. This ordering is similar to that of using these scores for standard data filtering, and we would expect some of the other proxies you’ve mentioned to be ordered similarly.
> 3. The additional preprocessing time is a linear pass over the dataset, and the additional memory required is minimal. This is because we can pre-calculate thresholds, and probabilistically sample, as suggested in our pseudocode (Listing 1). Typically the metrics used to rank documents have already been calculated; in our case, FastText scores are calculated during the creation of DCLM, while counts are calculated during deduplication.
> 4. Thank you for the suggestion.

---

### Official Review · Reviewer_fPkP · 2025-07-07

**Clarity:** 3
**Significance:** 3
**Originality:** 3
**Rating:** 5
**Confidence:** 4

**Summary:**

This work aims to study how best to use heavily filtered and deduplicated datasets at larger training scales. The first half of the paper considers repetition at the dataset level, building on works such as (Muennighoff et al. [19]). The second half considers repetition at the individual document level with the goal of creating better datasets given a token budget. The authors end by reflecting on the scenarios where building better datasets is likely to have the most impact.

**Questions:**

1. In Table 3 of Section 4, duplication count is used as or as part of the quality metric. Can the authors explain the motivation behind this?

2. Can the authors expand on the motivation behind the duplicate-aware sampling. The paper says this is done "to mimic the duplicate profile of the original dataset." Is this intended mainly as a comparison to the count manipulation methods which also have a high number of duplicates but perform much better (presumably because the documents were chosen according to quality).

3. In duplication-aware sampling, all or none of the duplicates are included probabilistically while for count manipulation the number of an duplicate is capped at some user chosen quality (the highest used in this paper is 5). Could this also be part of the difference between duplicate-aware subsampling and your count manipulation methods?

4. Line 102-103 says the following:

```
In Figure 2, we study a scenario similar to the findings in Muennighoff et al. [19], but instead start
with Chinchilla optimal unique tokens for 1B parameter models.
```

Are these tokens then excluded for the dateset subsets that are constructed for the rest of the experiments?

**Ethical Concerns:**

["NO or VERY MINOR ethics concerns only"]

**Final Justification:**

The paper is well-written and explores the interaction between data quality and repetition in a series of generally well-designed experiments. The results provide useful insights for practitioners when limited curated data becomes a constraint. In the rebuttal, the authors have updated the paper to better outline the flow of experiments and the clarity of the plots. I thus recommend acceptance.

**Limitations:**

Yes, the authors have adequately discussed the limitations.

**Quality:**

4

**Strengths And Weaknesses:**

**Strengths**

The paper is well-written and explores the interaction between data quality and repetition in a series of generally well-designed experiments. The results provide useful insights for practitioners when limited curated data becomes a constraint.

The first group of experiments in Section 3 mainly explore how the effects of repetition varies on 3 datasets (C4, a replication of RefinedWeb, and DCLM-baseline) across different scales:

1. The first experiment (Figures 2 and 3) explores how repetition affects performance for each of the 3 datasets at the 1B scale. Each run starts with training the model to the Chinchilla optimal duration with unique tokens from the datasets. Then 3 different separate runs are done with different size pools of the data; for each run the perplexity on the C4 validation set is reported as well as the averaged performance on a set of downstream tasks. For DCLM, the same experiment was performed at the 3B scale.

2. The second experiment (Figure 4) is similar except it fixes the total training budget across runs and then subsets the data to achieve the desired number of repetitions. This is done for different model sizes and durations on the DCLM-baseline data; each model size/duration also includes the baseline performance on one epoch of RW v2.

3. The third experiment (Table 1) explores using weight decay to mitigate the performance degradation from repeating tokens. Here the experiments are fixed at model size 12.6B parameters, duration 252B tokens. The number of repeats and magnitude of the weight decay is then varied. Figure 1 then emphasizes a scenario where weight decay can be used to meaningfully improve performance at the high repeat scale.

Section 4 then turns to how to "create better datasets relative to a token budget by explicitly manipulating the counts of individual document," focusing on repeats at the document rather than dataset level. Two main techniques are explored: subsampling methods and count manipulation. Count manipulation fills the token budget with documents in descending order according to some quality metric and utilizes different rules for how many documents to keep at each level of quality. The authors emphasize that count manipulation becomes fundamentally different from filtering in that high quality documents can be upsampled so that they appear more times in the new dataset than they do in the original. The performance of these strategies is explored in Tables 2-4.

**Weaknesses**

See the questions section for some concerns regarding Section 4 where I think the motivation behind some of the design choices could be made clearer.

Overall, some of the figure legends and captions could be improved to make the experiments easier to parse for the reader. A few specific examples (primarily to aid in improving the paper in revisions):

The caption for Figure 4 says "Legend denotes model size and multiplier to scale tokens seen at Chinchilla optimal." I presume this means that this is setting the total duration for each run in terms of a multiplier on the Chinchilla optimal duration, i.e. the duration for each run is the multiplier times 20x the number of parameters. However, because you are also changing the number of repeats this notation is not immediately clear. My suggestion would be to label the 6 colors with a more verbose description of the model size/duration settings and then have 2 more labels that explain that the solid and dashed lines correspond to the 2 datasets.

On the left hand side of Figure 1, since there are only 4 compute settings it would be useful to include the model size/training duration in the figure or caption. I assume from the $x$-axis label that the duration is the Chinchilla optimal for each model size, so just the model size could be sufficient. On the right hand side, it would help to clarify that 12.6B 252B refers to the model size and duration (later you have things like DCLM 230B where the number refers to the dataset size and not the duration). Also in Figures 1 and 2 you use the abbreviation RW v2 which isn't explained until page 4.

On the right hand side of Figure 1, it looks like "DCLM 12.6B 252B" is the WD 1x column from Table 1 and "DCLM 12.6B 252B WD" is the max performance for each row, which is mostly WD 2x. Some explanation of this would be helpful to make it clear why the curves match at 1 epoch. Also I presume that the largest compute scale in the left panel is the 12.6B parameter, 252B duration scale explored in the right panel. Using some marker to indicate that the 1 epoch and 10 epoch points have matching values on the left panel could be helpful given that the current shared colors are confusing (the 1 epoch point is coming from the purple line on the left).

---

> ### Author Rebuttal · Authors · 2025-07-31
>
> Thank you for your suggestions for improving our paper.
>
> We have updated our draft to take into account your suggestions regarding presentation.
>
> Regarding your questions:
> 1. We started with deduplication count because our exploration in this section is motivated by the observation that deduplication can hurt dataset quality. This implies that data points are of different quality and may need to be weighted in a different manner. We then combine it with FastText score because it is considered a better quality indicator than deduplication count.
> 2. Duplicate-aware sampling is most useful for running small scale experiments. Scaling up experiments can often assume new unique tokens of the same quality, but in works (e.g. in the DCLM paper) without an initial global deduplication, scaling back up is now expanding our dataset by adding additional repeats. Duplicate-aware sampling would be one way to design data pools for filtering experiments.
> 3. Yes, this is why duplicate-aware subsampling performs worse. Another way to view this is that there are less unique documents after duplicate-aware subsampling when compared to naive subsampling.
> 4. No, in figure 2, when we increase the token count (e.g. go from 28B to 56B unique tokens) we re-sample from the dataset, so there may be overlap between the 28B dataset and the 56B dataset. Here, “start” refers to the smallest dataset that we repeat with, and in contrast to Muennighoff et al. have much higher unique tokens per parameter.

---

> > ### Comment · Reviewer_fPkP · 2025-08-06
> > **Response to Rebuttal**
> >
> > I have read the other reviews as well as the authors' response. Based on the response to my questions, I would encourage the authors to more explicitly discuss their reasoning behind using duplicate count as a quality score and for performing duplicate-aware sampling. I also concur with the other reviewers that an early summary of the flow of experiments would be useful; the authors say this has been added in an updated draft. Overall, I continue to recommend acceptance.

---

### Official Review · Reviewer_t4Kk · 2025-07-09

**Clarity:** 3
**Significance:** 3
**Originality:** 3
**Rating:** 4
**Confidence:** 4

**Summary:**

This paper investigates the practical trade-offs between data quality and data quantity in the pre-training of large language models (LLMs), particularly in the context of scaling compute budgets where the availability of unique, high-quality data is becoming a constraint. The authors challenge the conventional wisdom of single-epoch training on unique data by exploring the efficacy of repeating smaller, more aggressively filtered datasets. The paper's core findings are twofold:
+ Dataset-level Repetition: The authors demonstrate that repeating a high-quality, aggressively filtered dataset (DCLM-baseline) for up to ten epochs can yield superior model performance compared to training for a single epoch on its ten-times-larger, less-filtered superset (RefinedWeb). This surprising result is shown to hold across various compute scales, provided that the training recipe is adapted—specifically, by increasing weight decay to counteract the effects of repetition.
+ Document-level Repetition: Moving beyond dataset-level epochs, the paper explores the idea that not all documents are created equal. It introduces "count manipulation," a more granular approach to dataset construction where the number of times each individual document appears in the training set is explicitly controlled based on a quality metric (e.g., a classifier score or the document's original duplication frequency). Experiments show that intelligently upsampling high-quality documents and downsampling low-quality ones can create better datasets for a fixed token budget than standard uniform subsampling or simple deduplication.

**Questions:**

See above.

**Ethical Concerns:**

["NO or VERY MINOR ethics concerns only"]

**Limitations:**

See above.

**Quality:**

3

**Strengths And Weaknesses:**

# Strengths
+ A Counter-intuitive and Foundational Insight on Dataset Repetition: The paper's most impactful contribution is the robust demonstration that repeating a smaller, aggressively filtered dataset for multiple epochs can outperform training for a single epoch on a significantly larger, but lower-quality, superset. This finding provides a clear, evidence-based strategy for navigating the increasingly critical issue of data scarcity and directly challenges the prevailing "single-epoch" training orthodoxy in the LLM community.

+ A Novel, Granular Approach to Data Composition: The work moves beyond coarse, dataset-level repetition to explore the more nuanced concept of document-level repetition. By introducing "count manipulation," the authors reframe dataset creation as a principled optimization problem, where the frequency of each document is explicitly controlled based on quality metrics. This represents a more powerful and flexible paradigm than simple filtering or uniform sampling.

+ A Practical Roadmap for Data-Constrained Scaling: Together, these insights offer a practical roadmap for training high-performance models when the volume of unique, high-quality data is a limiting factor. The paper forces a critical re-evaluation of long-held assumptions and suggests that the community may have prematurely dismissed the utility of multi-epoch training, a standard practice in other deep learning domains.

# Weaknesses
+ Uncertain Extrapolation to Frontier-Scale Training Regimes: The primary weakness of the study lies in the scale of its experiments. While substantial for academic research, the models are trained on token budgets in the hundreds of billions. This is approximately an order of magnitude smaller than the over 30 trillion token datasets now commonly used to train state-of-the-art frontier models.

+ The Question of Shifting Scaling Laws: The core uncertainty is whether the observed phenomena will extrapolate linearly to these much larger compute scales. It is plausible that scaling laws may shift, and the optimal trade-off between data quality, quantity, and repetition could be qualitatively different in a multi-trillion token regime. Therefore, it remains an important open question whether the benefits of heavily repeating a 1T token high-quality dataset would still hold against a single pass through a 15T token moderately-filtered corpus.

---

> ### Author Rebuttal · Authors · 2025-07-31
>
> Thank you for your review of our paper.
>
> We agree that our models do not scale to the same levels of data and compute as state-of-the-art frontier models. We also agree that a more extensive scaling analysis of repetition vs decreased-filtering is a useful future extension to this work.
>
> However, we also believe that, at a minimum, our existing results demonstrate that increased regularization and taking into account data filtering can support higher data repetition than was previously demonstrated (by Mueninghoff et al) at similar scales. We believe that these results are worth sharing more broadly with the community.
>
> We would guess that in the particular case of DCLM it would perform poorly when repeated to 10T+ tokens because it contains less than 1T unique tokens. A more effective study at that scale would require new datasets, which may lead to slightly different behaviors and comparisons. We hope researchers continue looking into data filtering in relation to data repetition, as well as adjacent topics like improving token yield.
>
> We would also be curious of the reviewer’s opinion on whether these things can be clarified with better scaling laws or improved metrics instead of training larger models with larger token budgets.

---

### Decision · Program_Chairs · 2025-09-17

**Decision:**

Accept (poster)

**Comment:**

The paper shows the aggressive data filtering and repetition can outperform training for an epoch with unfiltered larger dataset, when the training process, in particular, regularization, is adjusted. The authors also study document-level repetition.

The reviewers noted that the writing and presentation could be improved, and some key details, such as around regularization, should be more highlighted. The motivation behind some design choices could also be made clearer. Most of these concerns were addressed during the rebuttal. The reviewers also requested sensitivity analysis, which was provided during the rebuttal but only covers weight decay.

One of the primary concerns is that the experiments were conducted on token budgets in the hundreds of billions, an order of magnitude smaller than the datasets used for state-of-the-art frontier models. It remains unclear whether the observed benefits of data repetition will extrapolate to larger scales. However, the findings may still be relevant at smaller scale models used in academic research.

A review of older related work on data pruning, data difficulty is missing (e.g., Sorscher et al., Paul et al., Toneva et al., and many many others, some of which may not be applicable in the LLM training, but are readily extendible and deserve a discussion). Similarly, discussion of work on the effects of various direct and indirect regularization methods is missing.

Another concern of mine is that the natural baseline is missing: training for longer on unfiltered data with regularization, especially were the model scale is larger than the "optimal" data regime let's say, as suggested by scaling laws. From what I can tell, the experiments on the larger unfiltered data are with one epoch training.